# Mechanisms of Microbial Plant Protection and Control of Plant Viruses

**DOI:** 10.3390/plants11243449

**Published:** 2022-12-09

**Authors:** Lakshmaiah Manjunatha, Hosahatti Rajashekara, Leela Saisree Uppala, Dasannanamalige Siddesh Ambika, Balanagouda Patil, Kodegandlu Subbanna Shankarappa, Vishnu Sukumari Nath, Tiptur Rooplanaik Kavitha, Ajay Kumar Mishra

**Affiliations:** 1Division of Crop Protection, ICAR-Indian Institute of Horticultural Research (IIHR), Bengaluru 560089, Karnataka, India; 2Division of Crop Protection, ICAR-Directorate of Cashew Research (DCR), Dakshina Kannada 574202, Karnataka, India; 3Cranberry Station, East Wareham, University of Massachusetts, Amherst, MA 02538, USA; 4Department of Plant Pathology, College of Horticulture, University of Horticultural Sciences (Bagalkot), Bengaluru 560065, Karnataka, India; 5Department of Plant Pathology, University of Agricultural and Horticultural Sciences, Shivamogga 577255, Karnataka, India; 6Donald Danforth Plant Science Center, St. Louis, MO 63132, USA; 7Department of Plant Pathology, University of Agricultural Sciences, GKVK, Bengaluru 560065, Karnataka, India; 8Khalifa Centre for Genetic Engineering and Biotechnology, United Arab Emirates University, Al Ain P.O. Box 15551, United Arab Emirates

**Keywords:** antiviral, *Bacillus* spp. bacteria, bioagents, endophytes, resistance, plant virus

## Abstract

Plant viral diseases are major constraints causing significant yield losses worldwide in agricultural and horticultural crops. The commonly used methods cannot eliminate viral load in infected plants. Many unconventional methods are presently being employed to prevent viral infection; however, every time, these methods are not found promising. As a result, it is critical to identify the most promising and sustainable management strategies for economically important plant viral diseases. The genetic makeup of 90 percent of viral diseases constitutes a single-stranded RNA; the most promising way for management of any RNA viruses is through use ribonucleases. The scope of involving beneficial microbial organisms in the integrated management of viral diseases is of the utmost importance and is highly imperative. This review highlights the importance of prokaryotic plant growth-promoting rhizobacteria/endophytic bacteria, actinomycetes, and fungal organisms, as well as their possible mechanisms for suppressing viral infection in plants via cross-protection, ISR, and the accumulation of defensive enzymes, phenolic compounds, lipopeptides, protease, and RNase activity against plant virus infection.

## 1. Introduction

Eco-friendly crop disease management is the one of most important pre-requisites for ecological and sustainable farming in the 21st century, as many plant diseases caused by fungi, viral, and bacterial organisms pose major yield-limiting factors and affect the quality of produce in economically important crops. Among the biotic stresses, plant viruses cause severe epidemics in all major agricultural crops of economic importance, posing a severe threat to global food security. Plant viruses are known to cause nearly half (47%) of the emerging and re-emerging plant disease epidemics worldwide [1], and plant viruses cause approximately 30% of crop diseases [2], and among them, more than 80% of plant viruses have genome of RNA. More than twenty-five plant virus families are able to attack a wide host range globally, resulting in massive economic losses [3], and approximately 50% species of plant viruses causing disease in plants are intracellular parasites. Plant viral diseases are frequently emerging primarily due to the changing pattern of climatic variables, trading of commodity, and the plant viruses evolving more rapidly [4]. During the 1980s, approximately USD 15–20 billion loss was caused by plant viral diseases [5] and more than USD 30 billion in 2014 [6]. This also contributes to intensifying the global economic impact of plant virus disease [7]. 

There are currently very few options available for managing plant viral diseases in the field, since the application of insecticides and other toxic material inhibit the virus transmission by vectors is a desirable and unavoidable management strategy. As a result of the indiscriminate and excessive use of insecticides, insect vectors have developed resistance mechanisms against them. Furthermore, the probability of virus recombination has increased the chances of evolving highly aggressive viral strains and strains, which are capable of overcoming genetically induced resistance in plants by means of resistance breakdown. Chemical control measures are unsuccessful, as viruses are intracellular pathogens. The prophylactic measures include the removal of affected crop debris and using a greater number of applications of pesticides to reduce the population of insect vectors. 

The need of an hour to identify bioagents, with their beneficial activity, environmental safety, and a combination of diverse biocidal strains against major pathogens and pest with the activation of immune reactions in plants via by activation of specific signaling cascades, induced systemic resistance (ISR), and secondary metabolites in virus-infected plants, is of great interest. When compared to chemical pesticides and transgenic approaches, beneficial and heterogeneous groups, such as actinomycetes, endophytic microbes, plant growth-promoting bacteria, and fungal agents, can suppress viral activity and inhibit the egg-laying capacity of insect vectors [8,9,10,11,12], even though several researchers have stated the importance of beneficial microorganisms in protecting plants from pests and pathogens, including plant viral diseases [13,14,15].

## 2. Microorganisms Used in the Management of Plant Virus Diseases

### 2.1. Actinomycetes in Plant Virus Inhibition 

Actinomycetes are possible candidates for the production of secondary metabolite compounds, known as antibiotics, for their role as biocontrol agents and plant probiotics potential, due to their plant colonization and in situ antibiotic production [16,17,18]. Several researchers have reported the antiviral activity of actinomycetes, such as *Streptomyces ahygroscopicus* [17,19,20], *Streptomyces* sp. ZX01 [21], and *Streptomyces noursei* var. *xichangensis* [22], against *TMV* infection in tobacco. Actinomycetes inhibit the virus through a variety of mechanisms, including the activation of plant defense pathways and the production of signaling molecules. Marine organisms differ metabolically and physiologically from terrestrial habitants, and marine actinomycetes have been identified as a possible origin of numerous unique bioactive compounds [23]. Latake and Borkar [24] investigated the antiviral activity of metabolites from 28 marine actinomycete isolates against *Cucumber mosaic virus* (CMV).

In general, bioagents have diverse advantages in the reverence of little mammalian toxicity, biodegradability, superior ecological compatibility, and a distinctive mechanism of action. Metabolite of *Streptomyces olivaceus* was found to be impressive in controlling the CMV infections under in-vitro and open fields when applied individually as seed and spray treatment. A decreasing trend was observed in various necrotic lesions of *Tobacco mosaic virus* (TMV) and *Potato virus Y* (PVY) on *Nicotiana glutinosa* and *Chenopodium quinoa*, respectively, by application of the *Streptomyces* culture filtrates and virus sap mixture [25]. Similarly, Xing et al. [26] observed the inactivation of TMV in *Nicotiana glutinosa* and *N. tabacum* by mixing the different components, such as fermented actinomycetes broth, along with TMV sap, at different time intervals. The antiviral compound, ε-poly-l-lysine (ε-PL) of 3454–4352 Da, produced from *Streptomyces ahygroscopicus* has anti-TMV property [27]. 

### 2.2. Bacteria and Its Consortium against Plant Viruses

Monitoring the global biocontrol market revealed that there are no antiviral biopreparations in the biopesticides group that act directly as antiviral agents of biological origin in nature [28]. The different bacterial strains, such as *Pseudomonas aeruginosa*, *Burkholderia* sp. and *Bacillus* sp., were evaluated against the *Cotton leaf curl virus* (CLCuV) under artificial conditions by applying individually and in different combinations. The highest inhibition of CLCuV was observed in plants applied with a mixture of bacterial strains and only 0.4%, and the viral load was estimated in challenged plants, as compared to 74% in control plants. The principal component biplot analysis (PCA) revealed that a highly significantly correlation was found among the attributes, the viral load, and the incidence of disease [29].

In tomato seedlings, when subjected to the *Pseudomonas* sp. consortium of chitosan-based formulation, three demonstrated the increased effect of ISR and resulted in the accumulation of chitosan, which led to the enhancement of biocontrol efficacy against *Tomato leaf curl virus* (ToLCV) by application of *Pseudomonas* sp. The minimum viral titre was quantified through a semi-quantitative PCR assay with the application of chitosan and *Pseudomonas* sp. and scanning electron microscopy revealed a significantly higher number of bacterial cells in the roots, with no morphological or other qualitative differences [30]. *P. fluorescens* was discovered to have field efficacy for CMV and *Tomato mottle virus* (ToMoV) [31], biocontrol activity against *Tospovirus* [32] in *Solanum lycopersicum*, and lesion-inducing *Tobacco necrosis virus* (TNV) disease in *Nicotiana tabacum* resulted in a reduction in TNV-induced lesion number in *P. fluorescens* treated plants [33]. Zehnder et al. (1999) [34] identified plant growth-promoting rhizobacteria (PGPR) strains that protected tomato plants from systemic CMV infection. The main indirect use of PGPR is biocontrol of plant diseases. Generally, the major biocontrol activity of PGPR is by nutrients competition, niche elimination, metabolite production, ISR, etc. The bacterial bioagents were reported against the different plant viral inhibition reviewed by Maksimov et al. [11]. The efficacy of virus control also depends on the method, concentration, and time of application of the bacteria bioagents. The strains of several bacterial species viz., *Azotobacter vinelandii* and *Azotobacter chroococcum* [35], and *B. subtilis* Ch13 [36] inhibited the *Potato virus X*(PVX), PVY, and *Potato leaf roll virus* (PLRV) in *Solanum tuberosum*; *B. cereus* (I-35) and *Stenotrophomonas* sp. (II-10) reduced the infection of TMV, *Chili veinal mottle virus* (ChiVMV) in *Capsicum annuum* [37]; *Bacillus subtilis* 26D and *B. subtilis* Ttl2 against PVX and PVY [38]; *Bacillus* spp. against *Tobacco streak virus* (TSV) in cotton [39]; *B. amyloliquefaciens* MBI600 against *Tomato spotted wilt virus* (TSWV) in *S. lycopersicum* and *Solanum tuberosum* [36]; *B. amyloliquefaciens* FZB24, *B. pumilus* EN16, *B. subtilis* SW1, *Pseudomonas syringae* (heat-killed cells) [40], *P. putida* A3 [41], *Rhodopseudomonas palustris* GJ-22 against TMV in *N. tabacum* [42,43,44]; *Pseudomonas chlororaphis* O6N against TMV in *N. tabacum* cv Xanthi-nc [45]; *Bacillus pumilus* T4 and *B. subtilis* GBO3 against *Bean common mosaic virus* (BCMV) in *Vigna unguiculata* [46]; *B. pumilus* SE34, *B. amyloliquefaciens* 5B6, *B. pumilus* SE34, *Kluyvera cryocrescens* IN114, *B. amyloliquefaciens* IN937a, *B. subtilus* IN937b, *Pseudomonas lachrymans* against CMV in *Cucumis sativus* [47]; *S. lycopersicum* [48,49], *Capsicum annuum* [50]; *Paenibacillus lentimorbus* B-30488 against CMV in *N.tabacum* [51]; *Serratia marcescens* 90-166 against CMV in *A. thaliana* [52]; *Pseudozyma churashimaensis* against CMV, *Pepper mottle virus* (PepMoV), *Pepper mild mottle virus* (PMMoV), and *Broad bean wilt virus* (BBWV) in *Capsicum annuum* [53]; *P. fluorescens* CHA0 against *Urdbean leaf crinkle virus* (ULCV) in *Vigna mungo* [54]; *P. fluorescens* CHA0 [45], *P. fluorescens* P3 [33] against TNV in *N. tabacum*. 

Bacterial microbial consortia are also reported to control many viruses. The reduction of CMV infection was observed by using bacterial consortia viz., *B. subtilis* GB03 + *B. pumilus* SE34/*B. amyloliquefaciens* IN937a/*B. subtilis* IN937b/*B. pumilus* INR7/*B. pumilus* T4 in *S. lycopersicum* [55]; *Serratia marcescens* 90-166 + *P. putida* 89B-61/*B. pumilus* T4 against CMV in *Cucumis sativus* [48,56]; *B.licheniformis* MML2501 + *Bacillus* spp. MML2551 + *Pseudomonas aeruginosa* MML2212 + *Streptomyces fradiae* MML1042 against *Sunflower necrosis disease* (SND) caused by the *Tobacco streak virus* (TSV) in *Helianthus annuus* [57]; *B. amyloliquefaciens* IN937a + *B. pumilus* SE34 + *B. pumilus* T4 against *Papaya ringspot virus* (PRSV-W) and *Tomato chlorotic spot virus* (TCSV) by in *Carica papaya*/*S. lycopersicum* [58]; *P. fluorescens* Pf1. + *Bacillus* spp. EPB22 against *Banana bunchy top virus* (BBTV) in *Musa* spp. [59]. 

The combinations of individual bacterial isolate/consortium and chemical immunoregulators, such as *P. fluorescens* CHAO + chitin reduced the infection of BBTV in *Musa* spp [60]; *Pseudomonas* sp. 206(4) + B-15+JK-16+chitosan inhibited the ToLCV infection in *S. lycopersicum* [30]; *B. polymixa* + *P. fluorescens* + chitin were used against *Squash mosaic virus* (SqMV) control in *Cucumis sativus* [61]; *B. pumulus* INR7 + benzothiadiazole against CMV in *Capsicum annuum* [62] are also very effective in plant virus control. 

The concurrent infection of a single plant by a variety of pathogen is progressively more familiar as a host resistance modulator and pathogen evolution driver. In agro-ecosystems, plants are the target of a large number of pathogenic microorganisms, and co-infection could be regular, and as a result, it is important to consider. Co-infection was directed to raise bacterial specific symptoms, whereas there was a decrease in viral load, compared to the mono-infected plant. This could be due to gene silencing mechanisms intervening within plant interaction between virus and plant pathogenic bacteria. Therefore, pathogen–pathogen–host relations positively justify more consideration, from a hypothetical and practical point of view. A few of the co-infected bacteria in plant virus disease management with host plants and treatment methods in the reduction of virus infection are listed in Table 1.

### 2.3. Fungi in Plant Virus Inhibition 

Endophytes and fungal bioagents are able to recognize the changes in physiological means in stress-induced plants as defense machinery, thus regulating the plant gene expression [67,68]. Inoculation of cucurbits with *Colletotrichum legendarium* or TNV protects plants systemically against diseases caused by several pathogens. Muvea et al. [69,70] discovered that endophytic fungi (*Hypocrea lixii* F3ST1) inoculation on onion reduced the thrips vector population, resulting in higher death rate of vector population, and due to antixenotic repellence activity of the applied agents, reduced feeding behavior and oviposition could be observed. Furthermore, the reduced vector feeding activity of endophyte-colonized onions may decrease the virus spread of vectors. The endophytic interaction increases the incompetence of viruliferous thrips to transmit *Iris yellow spot virus* (IYSV) (Genus-*Tospovirus*), transmitted by *Thrips tabaci*, which has negative effects on IYSV replication in the infected plants. This may be due to the possibility of endophytes, such as fungi, eliciting the activation of gene expression in defense cascade pathways, in turn enhancing the accumulation of defensive compounds for development of resistance [71,72]. 

The influence of fungal secondary metabolites synthesized by endophytic association could be responsible in fungal-viral interplay mechanism. The alkaloids produced as a result of fungal endophytes application are found to have antiviral properties [73]. Endophytic colonization of onion seedlings may inhibit the feeding damage by viruliferous thrips. Furthermore, endophytic colonization also improves the futility of viruliferous thrips transmitting IYSV and has a negative impact on IYSV replication in the plant. As a result, endophytic fungi can be an essential component for tripartite (plant-endophyte-insect) interactions studies. Similar results were also reported in aphids, especially in *Rhopalosiphum padi* transmitting *Barley yellow dwarf virus* (BYDY), which showed a reduced aphid population and percentage of BYDV infections in fungal endophyte (*Neotyphodium uncinatum*) infected in meadow ryegrass (*Lolium pratensis*), in comparison to endophyte-free plants [74], inferring the production of alkaloids by endophytes, such as lolines, may help in fighting meadow ryegrass from BYDV infections [75,76].

A probable reason for the low virus load observed between the plants is most probable and likely to inhibit or down-regulate the coat protein gene expression, resulting in impaired virus replication during the initial phases of the infections. Peanut bud necrosis virus, for example, was also managed by blocking the systemic movement of the virus in wild *Arachis* by application of bioagents [77]. Virus replication and systemic spread by cell-to-cell movement might be interfered with by plant defense signaling, leading to ISR [78,79]. There are different biocontrol mechanisms, among which, ISR is the most effective in reducing the viral disease in infected host plants. The rhizosphere of many plant species is colonized by PGPR, which provides additional beneficial effects to the host plants, *viz*., enhanced plant growth vigor, and disease resistance caused by fungi, bacteria, nematodes, viruses, and viroid’s [80]. The fungal bioagents reported against the different plant viral disease are presented in Table 2.

### 2.4. Virus-Based Control of Plant Viruses

The virus load in infected plants can trigger for the release of secondary volatiles and other compounds, which are highly congenial for attracting vectors. When the vector tends to feed on the plant, the virus can produce an anti-feeding substance, causing the vector to flee to a new plant, as well as influence insect vectors to feed on healthy plants [98,99]. Viruses can also cause havoc among insects. Whiteflies that feed on TSWV infected plants grow slowly and have low fecundity [100]. Minor and mild strains of plant viruses can also act as elicitors in plants and it is reported in a *Pepino mosaic virus* (PepMV) of *S. lycopersicum*, which depends on the mild variants of PepMV for its induction [101]. The commercially available products in the market contain one or a combination of two mild virus strains.

## 3. Microbe-Induced Antiviral Mechanisms against Plant Virus Infection

### 3.1. Cross-Protection

Microbial biological agents control and protect crops against diseases through diverse mechanisms. The application of biological agents is an alternative option for pathogen control using cross-protection. Cross protection is a type of induced resistance that develops in plants against plant viruses. It is most effective among the closely related strains of the same virus, even though reports of viruses that are phylogenetically distinct protecting against each other exist. This phenomenon was described by the Dutchman Thung and the English man, Salaman, about seventy years ago, the prior immunization of host plants with a mild strain to defend against the challenge inoculation by aggressive or severe strain of the same plant virus, which leads to fits with future-proofing of production systems, known as mild strain cross-protection (MSCP) [102]. *Citrus tristeza virus* (CTV), belonging to the genus *Closterovirus* transmitted by aphid vectors, has been successfully controlled by cross-protection in several parts of the world [103]. 

The mechanisms of cross-protection are still unknown; however, several plausible mechanisms are proposed, such as antibody formation, specific adsorption by new cell compounds, exhaustion of essential metabolites, limited replication sites available to the plant virus multiplications, competitiveness between main and severe viruses for host components, and point of replication sites within the cells, interfering with disassembly, secondary virus translation, or replication [104,105,106] and aggravating RNA silencing by the protector virus that guides to a sequence-specific deprivation of the test virus RNA [107,108]. One of the commonly debated cross-protection mechanisms is RNA silencing. During the mechanism, Dicer-like (DCL) enzymes recognize the double-stranded RNA (dsRNA) formed during viral replication and are sliced into small fragments of 21–26 nucleotide length [109,110]. The small nucleotide fragments are recognized as “small-interfering RNAs” (siRNAs), which can escalate the RNA silencing in the plants by establishing an RNA-induced silencing complex (RISC) composed of Argonaute proteins. Complete silencing results when siRNAs are spread all over the plant, triggering the silencing of the virus to be stimulated in plant parts that had no prior viral interaction. During general silencing, RNAs are analogous in sequence to the RNA that first activated the silencing in all plant parts. Furthermore, the Argonaute proteins have also been concerned with the translational suppression of viral RNAs, stimulated by a mixture of viral eliciting agents and host resistance factors [111,112], and this needs more comprehensive research.

Additionally, *ZYMV* in squash, melon, and watermelon [113,114,115], *Cocoa swollen shoot virus* in cocoa [115], *Tomato mosaic virus* (ToMV) in *S. lycopersicum* and pepper [116], and *Papaya ringspot virus* in papaya [117] are some of the examples of viruses for which such claims were revealed to be effective. In the past two decades, a post-transcriptional gene-silencing (PTGS) method has been utilized to activate antiviral responses in plants by combining genome-editing methods, such as CRISPR/Cas9 and genetic transformation methods [118]. However, the mechanisms in antisense generally work for the RNA replication inside the nucleus in double-stranded DNA reverse transcriptase (dsDNA-RT) viruses (caulimo- and badna-viruses) [119]. For plant infecting RNA viruses’ control, ribozymes that can slice viral RNA can be used [120,121].

### 3.2. Antiviral/Antibiotic Compounds

Polysaccharides, polysaccharide peptides, and proteins are the main antiviral compounds existing in fungi. For the past few years, from fungal organisms such as *Coriolus Versicolor*, *Coprinus comatus*, *Lentinus edodes* (Berk.) sing, *Pleurotus ostreatus*, and *Flammulina velutiper*, polysaccharides and their peptides with antiviral abilities have been studied [122]. A polysaccharide peptide extracted from *Coriolus versicolor* showed 85.4% and 64.8% anti-TMV action lower concentrations [123]. The lentinan component exhibited an anti-TMV activity with a restorative percentage of 58.7% at a 10 μg mL^−1^ concentration [88]. The lentinan and polysaccharide peptide components might encourage the expression of peroxidase and phenylalanine ammonia-lyase and, similarly, that of the pathogenesis-related (PR) proteins in host plants, thus strengthening the plant’s inborn immunity to many diseases. 

Kulye et al. [90] reported numerous pathogenic fungi with low pathogenicity against plant viruses in which *Alternaria tenuissima* can successfully help the immunization of plants. A protein with higher capacity may induce plant immune responses, plant resistance, and growth metabolism. Other fungi having antiviral-active compounds include *Agrocybe aegerita*, *Flammulina velutipes*, *Lentinus edodes*, *Pleurotus citrinopileatus*, and *P. eryngii.* Additionally, *Neosartorya fischeri* and *Penicillium oxalicum* fungal methanolic extracts have shown repressive action to TMV infections [95]. The antimicrobial substances, such as peptaibols and trichokonins, of the fungus *Trichoderma pseudokoningii* SMF2, are an ISR determining factor that exhibited resistance against TMV [95]. Alkaloids are also revealed to have antiviral functions [124]. Similarly, *Trichoderma harzianum* has condensed TMV symptoms expression on *S. lycopersicum* through ISR activity [125].

The understanding of the effectiveness of bacteria and other microorganisms in improving growth of plants and inhibiting virus infectivity benefit the plant virus control in the field, particularly for those apprehensive about the eco-friendly control of crop diseases. The use of antiviral compounds that activate systemic resistance in plants has been described for many viruses in plants. Microbe-derived antiviral agents afford environmentally friendly, efficient, and degradable backup strategies for traditional chemical agents to control plant viral diseases. Their mechanisms against plant viruses are viable for directly acting on viral nucleic acids or proteins, or ultimately, restrain viruses by regulating host reaction [126]. 

The actinomycete group comprising 80 genera produces a variety of secondary metabolites with excessive inference in crop protection. More than 50% of the identified antibiotics are sourced from actinomycetes [127]. *Ningnanmycin* (NNM), an antiviral substance extracted from actinomycetes-*Strepcomces noursei* var. *xichangensis*, a new cytosine nucleoside peptide antibiotic, is most successful in inhibiting TMV infections, which is widely used in crop production [87]. It was well-defined that Ningnanmycin may be accountable for increased resistance against TMV-infected plants through triggering multiple plant defense pathways, induction of peroxidase (POD), phenylalanine ammonia-lyase (PAL) and superoxide dismutase and (SOD) activity, and activation of group of acidic PR proteins, and the expression of the NPR1and Jaz3 results in the preventive effect of TMV CP [22]. The NNM is a cytosine nucleoside type antibiotic that could increase the concentration of the Rubisco large subunit (Rubisco LSU) and Rubisco small subunit (Rubisco SSU), which, in turn, reduces the viral CP content. Rubisco is essential for carbon fixation in the plant system. NNM has a high ability to alleviate photosynthesis injury by suppressing the TMV CP inside chloroplasts, as CP might distress photosynthesis in virus-affected plants by inhibiting PS II activity [22].

The cytosine peptidemycin, extracted from *Streptomyces ahygroscopicus*, showed competent virucidal action [20], and it has been promoted, registered, and marketed as an anti-plant-viral agent in China. Additionally, Zhang et al. [21] reported a new glycoprotein GP-1 isolated from *Streptomyces* sp. ZX01, with an 80% anti-TMV rate at 1 mg mL^−1^ concentration. Peroxidase, is well-thought of as an authoritative defensive enzyme, is evidenced in numerous physiological responses against the biotic stresses of plants [128,129]. 

Galal [130] described that *Streptomyces* strains induced the systemic acquired resistance (SAR) to virus infections, whereas *P. aeruginosa* enhanced the resistance to TMV in tobacco plants [131], as well as an antiviral compound from *S. Noursei var xichangensis*, known to induce ISR [22]. Li et al. [132] reported that *S. pactum* Act12 ISR against TYLCV and salicylic and jasmonic acid concentrations have amplified in *S. lycopersicum* plants. The various bioactive compounds derived from *Streptomyces* strains were found to be effective in minimizing the TMV local lesions on the leaves of Datura *metel* weed plant [133]. Additionally, a bioactive compound defined as the ε-Poly-l-lysine, released from by *S. ahygroscopicus*, has revealed substantial defense and curative action against TMV [27]. The mosaic symptoms produced by the ZYMV has been shown to change percent reduction to 95% and 100%, with the foliar spray on *Cucumis sativus* with *S. Albovinaceus* and *S. sparsogenes,* separately [134]. The use of antibiotics in plant pathology, particularly in plant virology, has long been assumed to be of critical importance. *Cytovirin* is a broad-spectrum antibiotic that reduces virus concentration in the host and delays symptom emergence [135]; ACD (*Actinomycin* D) protects soybeans from *Bean pod mottle virus* (BPMV) [136]. TMV inhibition has been accomplished through the use of blasticidin S, dextromycin and mitomycin C [137], oxytetracycline and streptomycin [138], puromycin [139], laurisin or formycin [140], cycloheximide [141], chloramphenicol [142], etc. *Pea streak virus* (PSV) may be controlled by spraying cycloheximide [143], actidione and streptomycin [144], and blasticidin S [145]. Phatak and Batra [146] reported the control of *Sunn-hemp mosaic virus* (SHMV) in leguminous plants by using pentaene G 8, an antifungal antibiotic from *Streptomyces anandii*. Leaf hoppers were unable to pick up viruses when the host plants were applied with tetracyclines [147]. *Bromegrass mosaic virus* has been controlled by actinomycin D and blasticidin S [148]. Root dipping and foliar spray have been recommended for ohyamycin and blasticidin S against PVX, and tomato leaf curl may be effectively handled with DPB [149,150], and this antibiotic also helps in yield increase in *S. lycopersicum*. The bacterial bioagent *Bacillus amyloliquefaciens* (VB7) proficiently reduced the expression of GBNV disease symptom up to 84% through the transient expression of MAMP genes, triggered immune reaction, and decreased virus titer [151].

In some of the reports, the treatment of antibiotics to virus-infected plants leads to reduced symptom expression, besides the inhibition of plant viruses [152,153]. Still, the prominent actions of antibiotics on human pathogens have not been established in the framework of plant viruses. Amongst the few stated antibiotic activities of chloramphenicol, it principally interferes in protein synthesis [154]. Similarly, the antibiotic daunomycin also has effects in few protein synthesis steps and prevents nucleic acid metabolism [155], similar to actinomycin D. However, daunomycin had only a negligible consequence on virus-infected plants, either on viral RNA synthesis or on the host plant [156]. They thought that this antibiotic had a straight effect on viral particles.

Actinomycin D and mitomycin C antibiotics are also identified to affect the metabolism of nucleic acid and form complexes with DNA. Excluding in the very early stages of infection with specific virus-host amalgamations, they do not obstruct the plant pathogenic RNA virus synthesis [157,158]. CMV may be suppressed by miharamycin A, an antibiotic, which is known to prevent DNA-dependent RNA synthesis [159]. It is anticipated that the antibiotics are unified into the host metabolic pool based on their molecular arrangement, inhibiting the virus from appropriately intermingling with the host metabolic activity. As foliar sprays, antibiotics cause irregular virus propagation or mislead the synthesis of coat protein [160]. Antibiotic inhibition of virus-RNA synthesis was studied using virus-infected leaf discs floated in antibiotic solutions [161]. Off late molecular virologists and phytopathologists are equally fascinated by this field.

The algae group contains a diversity of plants ranges, from diatoms observed as microscopic and unicellular to seaweeds spreading over 30 m. The first discovered sources of natural compounds are microalgae against HIV [162]. The polysaccharide of seaweeds, exactly sulfated polysaccharides, from brown algae has effective virucidal properties [163,164]. Alginate, a profuse polysaccharide from brown algae, exhibited 95% inhibition of PVX at 10 mg mL^−1^ concentration [164]. From the methanolic extracts of 30 species of marine algae, only six species showed more than 80% inhibition rates against PVX at 10 mg mL^−1^ concentration. Nagorskaia et al. [165] found that red marine alga, *Tichocarpus crinitus*, released kappa/β-carrageenan that can decrease TMV infection in Xanthi-NC tobacco leaves, and the lectin from the marine algae, *Ulva pertusa*, have anti-TMV activity [166].

Animals have an insufficient number of anti-plant-virus compounds than plants and microbes. Amongst them, oligochitosan is the most successful anti-plant-virus compound produced by enzymatic hydrolysis of chitosan polymer discovered in animals. Chitin and chitosan have been discovered to be nontoxic, biodegradable, and biocompatible substances that stimulate the broad-spectrum defensive response in plants [167,168]. The chitosan is deacetylation product of chitin and has been shown to have virus destruction irrespective of plant species, as well as the virus type [169]. An oligochitosan at 50 μg mL^−1^ concentration inhibits 50.41% TMV infection. Since then, oligosaccharide has been marketed as a vital anti-plant virus compound in China. Similarly, melittin and its analogue and chondroitin sulfate, a whey protein, and its esterification products were identified as anti-plant virus compounds [170,171,172,173]. Several researchers have found that oligochitosan interferes in pathogen infection through stimulating the production of hydrogen peroxide, nitric oxide, protein kinase, Ca2+ signaling pathway, and promoting PAL activity [174,175,176,177].

### 3.3. Lipopeptides

Lipopeptides are antimicrobial peptides synthesized by a wide range of microorganisms via non-ribosomal synthesis, including secondary metabolites, such as peptaibols, cyclopeptides, and pseudo peptides [178]. Many researchers have screened bacterial agents for their virucidal activity and isolated bioactive compounds responsible for the inhibition of plant viruses. Zhou et al. [179] reported that the bacterial strain ZH14 produces bioactive proteins with durable and stable TMV resistance. Shen et al. [180] defined the improved control efficacy of *P. fluorescens* CZ against TMV. Thapa et al. [181] described that *Serratia marcescens* culture filtrate has stable antiviral capacity against CMV. Cell-free supernatant consisting of cyclic peptide synthesized from *Pseudomonas chlororaphis* O6 exhibited strong antiviral activity against TMV [45].

Lipopeptides are 1000–2000 Da cyclic, lower antimicrobial compounds that are primarily synthesized by bacteria, such as *Bacillus* and *Pseudomonas* sp. [182]. Indeed, the lipopeptides are the utmost critical factors providing for their biocontrol efficacy in the PGPMs. Lipopeptides are synthesized through a multi-enzyme biosynthesis pathway with a precursor of specific gene clusters from non-ribosomal peptides synthetase (NRPs) [178]. Additionally, numerous lipopeptides viz., iturin, surfactin, sophoro lipids, trehalose lipid, rhamno lipids, and mannosylerythritol lipids displayed promising effective substitutes of biocontrol agents for many applications in agricultural production [183].

The *Bacillus* spp. produces surfactin, iturin, and fengycin of three major families containing hydrophilic amino acids (7–10 amino acids) connected with a hydrophobic fatty acid tail. These compounds perform as effective antagonists by inhibiting plant pathogens by diverse mechanisms by inducing and establishing the plant defense apparatus in defense-related systems recognized as ISR [184]. Of late, some of the cyclic lipopeptides have been recognized as elicitors of plant defense response, such as ISR. The volatile compounds, such as 2, 3-butanediol [52], released by *Bacillus* spp were evidenced as elicitors to activate ISR in plants. Lipopeptide surfactants exhibit an exceptional aperture and ion networks establishing possessions and, thus, may interrupt the normal integrity and absorptivity of the lipid layer of the plasma membrane. Lipopeptides, by virtue of their capability to interrupt the physical integrity of the living membrane, create their primary means of anti-microbial activity against diverse microbes, such as bacteria, fungi, viruses, mycoplasma, etc. [185].

*B. amyloliquefaciens* S499 releases cyclic lipopeptides and affords an efficient resistance in both the leaves and roots of sugar beet, consequential in biocontrol rhizomania disease through the reduction of the virus vector *P. betae* infection [186]. *Beet necrotic yellow vein virus* (BNYVV) causes rhizomania disease transmitted by the obligate root-infecting parasite *Polymyxa betae* [187]. *Bacillus* Cyclic lipopeptides (CLPs) significantly decrease the infection by *P. betae* in sugar beet through ISR. McGrann et al. [187] and Barr et al. [188] revealed the incomplete resistance to *P. betae* in sugar beet with decreased virus concentration. It seems that CLPs offers a new technique for the sustainable management of rhizomania disease.

The peptide compound indolil acetic acid and non-proteinogenic amino acid, such as the 5-aminolevulinic acid produced in *N. tabacum* plants treated with *Rhodopseudomonas palustris* GJ-22 strain, reduced TMV infection by stimulating the salicylate (NbPR1a and NbPR5) and jasmonate-mediated (NbPR3 and NbPDF1.2) signaling pathways genes [189]. However, the treatment with the *B. amyloliquefaciens* MBI600 strain induced host resistance to PVY and TSWV in *S. lycopersicum* by the expression of SA-induced signaling pathway genes [36].

### 3.4. Ribonucleases against Plant RNA Viruses

*Ribonucleases* are a group of hydrolytic enzymes that have the ability to dimerize and catalyze the breakdown of ribonucleic acid (RNA) into smaller units in several critical transcription phases and inhibit virus reproduction. *Ribonucleases* are the classical components of immune system [190]. Various enzymes, as ribonucleases, such as binases, baRNases, and baliphases, synthesized by the bacteria, have the ability to exude into the external environment. Small concentrations of ribonucleases help in plant growth, and and high concentrations of ribonucleases have antiviral properties and degrades viral RNA. In buckwheat cultivars, Kara-Dag and Roksolana displayed diverse resistance level of *Buckwheat burn virus* (BBV) by the expression of enhanced ribonuclease activity [191]. The expression of pac1 ribonuclease gene from *Schizosaccharomyces pombe* has improved resistance to CMV, ToMV, PVY, and TSWV in chrysanthemum, tobacco, and impatiens plants [192,193,194], even though inducing resistance to viruses by the expression of a double-strand-specific RNase gene altered the resistance levels from a delay in disease symptoms appearance to the complete protection against viruses. The expression of bacterial double strand-specific RNase gene is reported to show resistance to some viruses in tobacco plants [195]. Endophytic strains having RNases producing ability inhibit the spread of viruses and affects expression of viral symptom. High endophytic rates and RNase activity bacterial strains, such as *Bacillus* sp. TS2 and *B. subtilis* 26D, are used for the development of biocontrol agents. Several viral infections (for example, widespread PVS + PVY and PVM + PVY joint infections) hasten plant impairment substantially, compared to a sole virus infection, which was discovered by Hameed et al. [196]. Sorokan et al. [197] found a higher concentration of RNase activity in the culture media of *B. thuringiensis* B-6066, *Bacillus* sp. TS2, *Bacillus* sp. STL-7, and *B. subtilis* 26D, as well as strains with the enormous capacity to colonize internal plant tissues collective with increased RNase activity and decreased viral disease incidence and severity of potato viruses M, S, and Y. They distinguished that *Bacillus* spp. lessened the *Leptinotarsa decemlineata* egg clusters and larvae number and revealed their antifeedant activity on treated plants.

The bacterial species *viz*., *B. pumilus, B. amyloliquefaciens,* and *B. licheniformis* (binases, baRNases, and baliphases, respectively) have extracellular RNase secretive capacity to utilize organic phosphates in aiding bacterial adaptation mechanisms to changing ecological situations and slice RNA containing viral particles continuously in plant tissues [198,199]. Less concentration of RNases trigger plant growth and resistance to a diverse stress element, whereas high concentrations of RNases show antiviral activity by degrading RNA containing viral particles. Thus, the bacterial genera viz., *Pantoea, Cronobacter*, *Microbacterium*, and *Staphylococcus* isolates, which originated from the *Cucurbitaceae* family, produce nucleases, which cleave viral nucleic acids [200]. The viral particles in the juice from TMV-infected tobacco plants were shown to cleaved by *Pseudomonas putida* A3 [41] and *B. pumilus* 7P/3-19 [44].

There was a strong and stable constructive association between the RNase activity in several potato cultivars and their resistance to PVX, PVY, PVM, and PVS [192]. The RNase gene PAC1 expression from *Schizosaccharomyces pombe* in soybean crop caused a significantly higher concentration of *Soybean mosaic virus* (SMV) in SC3 strain-free soybean plants [201]. The *Nicotiana benthamiana* plants holding CRISPR/Cas13 a cassette that included class 2 type VI-A RNase, created by genetic engineering capable of identifying and slicing ssRNA, was extremely resistant to *Turnip mosaic virus* (TuMV) [202]. Approximately, one-third of the transgenic tobacco clones expressing the baRNase gene from *B. amyloliquefaciens* showed complete resistance to ToLCV infection [203]. Thus, these studies confirmed that *Bacillus* and other bacteria can protect plants against viral infection by upsetting vectors of viral particles, such as insects and plant pathogens.

### 3.5. Plant Growth-Promoting Rhizobacteria-Induced Resistance in Plants

The PGPRs are endophtic or rhizospheric bacteria which enhance yield sustainability, and growth promotion and defend plants from invading pathogens [204]. As a part of synergistic effect, PGPR help other microbes to improve their plant growth promotion or suppress pathogens. The compounds produced by plant growth-promoting microorganism (PGPM) may intercommunicate with the immune system of hosts and induce systemic resistance in crop plants against phytopathogens [205,206]. In general, PAMPs such as flagellin and lipopeptides of the endophytic bacteria [207] or viruses CP [208,209], are recognized by the receptors comprising leucine-rich repeats (LRR) [210]. The genes responsible for pathogenesis-related (PR) proteins, such as PR-4 and PR-10 with antiviral property expressed in the plants under rhizobacteria stimulation and their metabolic compounds, as well as in response to fungal viral [211,212] and infections. Thus, PGPM can play a major role in major plant reactions to viral infection.

The presence of bioagents/endophytes in plants recognizes the plant virus proteins, particularly viral RNA, as well as Coat protein (CP) of a virus, and the small conserved molecular motifs present in microbes called Pathogen-associated molecular patterns (PAMPs) are recognized by the receptors of plant cell, which results in the development of plant defense reactions [213] with rapid generation of Reactive oxygen species (ROS), changes in the phytohormone composition, and synthesis of different metabolites, including the regulation of defensive genes during local and systemic expression [214]. Li et al. [8] supported that *Enterobacter asburiae* activated the TYLCV resistance and lower disease rates, approaching 42%, even at 45 dpi, under greenhouse conditions. Regulation of disease resistance was detected against ToMoV in *S. lycopersicum* plants by the different strains of *B. amyloliquefaciens* [31].

Some of the rhizobacteria interacting with the plant roots result in a resistance to bacteria, fungi, and viruses, causing plant diseases, and this phenomenon is called ISR. In addition, ISR also involves jasmonate and ethylene signaling, which stimulates the defense responses in the host plant against diverse range plant pathogens [215]. PGPRs primarily protect the plants by CMV infection by inducing the natural resistance against the invading pathogen [31]. The PGPRs induce systemic resistance either through the ethylene and jasmonic acid ISR pathway or the salicylic acid-dependent SAR pathway [204,216]. Hypersensitive response (HR) is characterized by localized necrosis and disruption of the virus’s systemic spread in plants. In HR reactions, necrotic lesions or ringspots are formed at the infection site on leaves, stems, and fruits, holding the phytopathogen within it, hence shielding uninfected tissues. HR occurs due to modifications in plant cell wall with increase in the concentration of superoxide and nitric oxide radicals, calcium ions, and increases in SA, JA, and H_2_O_2_ levels. HR reactions against PVYC and PVYO strains of PVY are regulated by potato *Nytbr* and *Nctbr* genes, respectively, in *Solanum tuberosum* crop [217]. The avirulence factor of the PVY virus is the helper component proteinase (HC-Pro) cistron of PVY, and *Nx*-mediated hypersensitivity and *Rx*-mediated resistance were produced by various coat protein (CP) subunits of PVX [218]. This mechanism does not seem to be related with RNA interference, which is also a chief approach for defending plants against RNA-containing viruses [219]. The application of chitosan and PGPR displayed the hidden potential for inducing plant resistance [220,221]. These beneficial bacteria may inhibit the virus infection by induced local acquired systemic resistance (LASR) through the production of phenolics, secondary metabolites, such as phytoalexins, salicylic acid, PR proteins (chitinase and β-1, 3-glucanase), a cell walls lignification, and callose synthesis [222]. The two endophytic bacterial strains *B. subtilis* 26D and *B. subtilis* Ttl2, secreted ribonucleases and phytohormones, inhibited PVX and PVY in *S. lycopersicum* plants. Both the 26D and Ttl2 strains triggered ISR by activating the transcriptional genes related to salicylate and jasmonate-dependent reaction and increased the content of cytokinins and decreased the level of indolacetic acid in PVX- or PVY-infected plants [39]. Polyphenolic substances are secondary metabolites, and they play a pivotal role in inducing plant promotion and defending against many biotic and abiotic stresses. Many PR-proteins play a significant role in antimicrobial activity against phytopathogens [223]. The production of PR protein in plants is stimulated by the infection of bacteria, fungi, viruses, or viroids [224,225,226]. In the phenylpropanoid pathway, PAL is the first enzyme involved in salicylic acid biosynthesis [189]. The infections of pathogens stimulate the SA, which is generally correlated with the increased PR-1 as a SA marker gene [227]. *B. amyloliquefaciens* 5B6 treatment with pepper plants reduced the CMV incidence by induction of transcription-encoding genes, such as PR-4, PR-5, and PR-10 proteins [50].

An authoritative PR protein or defense protein, such as the peroxidase gene, is involved in various useful responses against plant biotic stresses, along with studying pollutant degradation and management [33,129]. Galal [130] detailed that application of *Streptomyces* strains stimulated SAR for virus infections, while *P. Aeruginosa* was extremely potent at enhancing TMV resistance in tobacco [131]. Furthermore, an antiviral compound from *S. Nourseivarxi changes* has been shown to induce SR against TMV [22]. The bacterial-derived 2,3-butanediol has established a defense reaction against CMV and TMV by the accumulation of transcription of many defense marker genes viz., *Capsicum annuum* pathogenesis-related 4 (CaPR4), Ca chitinase 2 (CaChi2), CaSAR8.2, Ca phenylalanine-I ammonia-lyase (CaPAL), Ca 1-aminocyclopropane-1-carboxylic acid oxidase (CaACC), and Ca proteinase inhibitor 2 (CaPIN2), which was similar to the increase in genes expression in plants received benzothiadiazole [228].

The rhizospheric *P. fluorescens* Pf1 and endophytic *Bacillus* spp. EPB22 bacterial mixture treated banana plants stimulated the defensive enzymes, such as peroxidase (PO), polyphenol oxidase (PPO), and PAL, in addition to the phenolic substances, which reduced the BBTV incidence up to 80% [59]. Similar responses were also detected in the BBTV-infected banana plants [63], TSWV-infected *S. lycopersicum* plants [229], and *Urdbean leaf crinkle virus* (ULCV)-infected black gram [54].

The cell wall glycoprotein of fungal phytopathogen, *Cladosporium herbarum* is peptidogalactomanann (pGM), induces defense-related genes for the expression of SAR and ROS and its accumulation with BY2 tobacco cells responsible for weakening viral infection [230], with enhanced expression of PR-1a (unknown function), PR-2(-1-3 endoglucanase), PR-3 (chitinase), PR-5 (thaumatin-like protein), PAL (phenylpropanoid pathway gene), and genes associated with plant stress responses and innate resistance for instance LOX1 (lipoxygenase) and NtPrxN1 (peroxidase) [230]. Sindelarova and Sindecor [231] reported that two PR proteins (PR-2a and PR-3) from *N. tabacum* displayed strong and durable antiviral action to TMV. Additionally, defense enzymes, such as peroxidase and PAL transcripts, showed strong antiviral activity [150,206,232].

Harish [233] reported that the application of rhizobacterial mixtures containing endophytic *Bacillus* spp. and *P. fluorescens* (Pf1) inhibit the BBTV. Chitosan and PGPR had reduced disease severity and *Squash mosaic virus* (SqMV) titre, which infects *Cucumis sativus* plants by deferring the incubation period in the reproductive phase (4-7 weeks after planting) [61] by triggering biochemical defense response. Additionally, the PGPR strains GB03 (*B.subtilis*) and IN937a (*B. amyloliquefaciens*) application with the chitosan as the carrier was effective against the CMV in *S. lycopersicum* plants [31]. Maurhofer et al. [33] detected that a few chitinases iso-forms in *P. fluorescens*-treated tobacco plants controlling the TNV and salicyclic acid production by rhizobacteria are also responsible for ISR against TNV. Pyung-II et al. [234] reported the induction of PR-1a and PAL in *P. fluorescens* strain EXTN-1 treated tobacco plants treated against the *Pepper mild mottle virus* (PMMoV).

The *P. fluorescens* strains are proficient in inducing significant levels of defensive enzymes in banana that *its induced* enzyme actions are linked with the biosynthesis of phenolics and other secondary metabolic compounds, which have been projected to be major factors in ISR against the BBTV disease along with higher yield [61]. Chirkov et al. [235] defined the connection of callose, β-1,3 glucanase, and ribonuclease induction as a defense reaction against PVX upon chitosan (1 mg·L^−1^) inoculation. The ALMV produced local infections on bean leaves (were fully controlled with the treatment of the maximum chitosan concentration (0.1%) through spray or addition to the virus inoculum [236]. Similar results were also described with the PVX, TMV, TNV, ALMV, PSV, and CMV [237,238]. Raupach et al. [56] were the first to demonstrate the treatment of *C. sativus* or *S. lycopersicum* plants with PGPR lead to ISR against the systemic infection of CMV.

The roles of ROS in plant–virus interactions are not well-understood. It is anticipated that ROS can act as a defense substance [43,238] and H_2_O_2_ as a systemic antiviral signaling unit during TMV infection. A higher amount of ROS production is also considered a biochemical marker during SAR induction.

The *Pepper leaf curl virus* (PepLCV) transmitted whiteflies are controlled by *T. harzianum*, *T. Polysporum*, *T. Atroviridae*, and their consortia. They occupy the pepper plants endophytically, with a considerable increase in the phenolic content (183% more), and induce innate host immunity by the activated phenylpropanoid biosynthesis. The competence of *Trichoderma* bioagents to produce salicylic acid seems to have a prominent role in composing the PepLCV suppression (up to 50%) and ROS accumulation at the infection point resulting in the restricted virus spread [81]. Siddique et al. [239] also reported considerable levels of higher phenolic content in the *Cotton leaf curl Burewala virus*-resistant genotypes than in susceptible ones after the inoculation. Abo-Zaid et al. [238] found that the foliar application of *Streptomyces cellulosae* (isolate Actino 48) at 2 × 10^7^ CFU mL^−1^ reduced the incidence of TMV in *S. lycopersicum* through ISR. They applied Actino 48 before TMV inoculation (48 h) and reported significantly increased levels of total phenolic compounds, proteins, peroxidase, and chitinase enzymes in TMV-treated tomato plants + Actino 48, as compared to TMV-treated tomato plants alone. Hence, Actino 48 could be used for the biological management of TMV. Due to the systemic nature of virus infection, effective chemical compounds cannot be applied for the control of plant viral disease in agricultural or horticultural crops. Although, the endophytic plant-growth-promoting bacteria (ePGPB) strains viz., *Paraburkholderia fungorum* R8, *Paenibacillus pasadenensis* R16, *Pantoea agglomerans* 255-7, *Pseudomonas syringae* 260-02, and chitosan-treated plants exhibited precise biocontrol activity against *Cymbidium Ring Spot Virus* (CymRSV) and CMV through a considerable decline in severity of virus symptom with increased plant height compared to the control. Furthermore, defense-related genes, such as enhanced disease susceptibility-1 (*EDS1*) gene up regulation indicated the involvement in salicylic acid (SA) signaling pathway, non-expressor of pathogenesis-related genes-1 (*NPR1*) involved in mediation of cross talk between SA or jasmonic acid (JA) and ethylene (ET) signaling pathways and induction of SAR in plant system by the activation of *PR2B*, a PR- protein results in activation of defense against virus infection [240]. The ePGPB protect crops many pathogens including virus by providing nutrients, plant hormones, and secreting allelochemicals and indirect biocontrol by exciting ISR in the host plant, which intern activates defense-related genes through the mediation of jasmonic acid, ethylene, and salicylic acid metabolic pathways. Biological control with PGPR microbes can be recommended to protect from viral pathogens, since PGPRs have direct and indirect roles in the sustainable management of crop plants through improvement of seed germination and emergence, plant growth promotion, biological nitrogen fixation, solubilization of phosphates, enhanced yield, yield components, and nutrient uptake, triggering ISR and other defensive compounds and enzymes, which are essential for disease resistance activity.

## 4. Future Prospects

Humans began to consider alternatives as a result of the detrimental effects of synthetic pesticides [241]. Pesticides can be replaced by biopesticides [242]. Diverse biologically derived compounds have pesticidal activity against insect pests and diseases [243,244]. It is imperative to evaluate the amount, number of applications, and suitable delivery approaches of these potential BCAs in field conditions for the effective management of plant viral diseases. Molecular shreds of evidence or the involvement of various defense or regulatory genes in combating many viral diseases are yet to be explored to the maximum potential. The defense-inducing mechanism of the bacteria, fungi, actinomycetes, and algae against plant virus infection and its low virus load in infected plants needs to be deciphered. The antiviral-inducing microbial populations need to be studied, concerning the optimum population of microbes, the time required for reducing the virus inoculum, and the effect of vector population with the virus on antiviral properties for effective management of plant viruses (245). Until now, there is no such precise and advanced study on the impact of diverse ecological features, such as rainfall, relative humidity (RH), temperature, and light hours, on interactions with antagonistic microbes, which is highly challenging in the induction of an antiviral resistance mechanism. The molecular mechanism of antiviral resistance induced by microbial agents is lacking and desires comprehensive revisions related to proteomics and metabolomics to unravel the plant microbial elements responsible for antiviral defense resistance.

## 5. Conclusions and Way Forward

Presently, numerous reports are available on the role of bacteria, fungi, actinomycetes, and other organisms in the involvement of defense mechanisms against plant pathogens, except for viruses. Basically, induced defense mechanisms against viral infection by microbes and their metabolites are impeding virus transmission and replication. It is suggested that the predictions of the exploitation of bacteria and plant RNase for the prevention of virus infection in plants are challenging and unexplored research areas. Thus, the identification of environmentally safe biological agents with antiviral properties for plant protection against virus diseases is a constructive method of plant defense. PGPRs have direct antiviral properties by generating RNases or SR-inducers, which live on surfaces or internal plant tissues, and such microbes indirectly decrease the viral load in the agro-ecological system through vector control by the “RNA biocides” specific for crop pests.

## Figures and Tables

**Table 1 plants-11-03449-t001:** Evidence of co-infected bacteria in plant virus management.

Bioagents	Plant Virus Control	Host Plants	Treatment Method	References
*Xanthomonas oryzae*	*Rice yellow mottle virus* (RYMV)	*Oryza sativa* L.	Foliar spray	[63]
*Erwinia tracheiphila*	*Zucchini yellow mosaic virus* (ZYMV)	Cucurbits crops	Foliar spray	[64]
*Bacillus* spp combinations	CMV	*Arabidopsis thaliana* L., *S. lycopersicum* L.	Foliar spray	[31,52,65]
*P. fluorescens*, *P. aeruginosa*	TNV	*N. tabacum* L.	Foliar spray	[33]
*Multiple rhizobacteria*	TMV	*Capsicum frutescens* L.	Soil drench	[66]

**Table 2 plants-11-03449-t002:** Antiviral inducing fungal microorganisms for the control of plant viruses.

Name of Fungal Bioagent	Virus Inhibition	Host Plants	References
*Hypocrea lixii*	IYSV	*Allium cepa* L.	[70]
*T. harzianum*, *T. Polysporum* and *T. atroviridae*	PepLCV	*Capsicum annum* L.	[81]
*Paecilomyces variotii*	PVX and TMV	*Nicotiana benthamiana* L. and *N. tabacum* L.	[82]
*Neotyphodium uncinatum*	BYDV	*Festuca pratensis* L.	[76]
*Beauveria bassiana (Balsamo Criv.)*	ZYMV	*Cucurbita pepo* L.	[83]
*Penicillium simplicium (GP17-2) (Trichocomaceae: Penicillium)*	CMV	*Arabidopsis thaliana* L. and *N. tabacum* L.	[84]
*T. harzianum and M. anisopliae*	*Sugarcane mosaic virus* (SCMV)	*Zea mays* L.	[85]
*T. harzianum*	CMV	*S. lycopersicum* L.	[86]
*Coriolus versicolor*	TMV	*N. tabacum* L.	[87]
*Lentinus edodes*	TMV	*N. tabacum* L.	[88]
*Agrocybe eaegerita*	TMV	*N. tabacum* L.	[89]
*Alternaria tenuissima*	TMV	*N. tabacum* L.	[90]
*Pleurotus eryngii*	TMV	*N. tabacum* L.	[91,92]
*Pleurotus ostreatus*	TMV	*N. tabacum* L.	[93]
*Pleurotus citrinopileatus*	TMV	*N. tabacum* L.	[91,92]
*Trichoderma pseudokoningii SMF2*	TMV	*N. tabacum* L.	[94]
*Penicillium oxalicum*	TMV	*N. tabacum* L.	[95]
*Coprinus comatus*	TMV	*N. tabacum* L.	[96,97]
*Flammulina velutipes*	TMV	*N. tabacum* L.	[91,92]
*Flammulina velutiper (Fr.) Sing*	TMV	*N. tabacum* L.	[93]

## Data Availability

Not applicable.

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
