# Peer review of "Mechanisms of Microbial Plant Protection and Control of Plant Viruses"

_plants, 2022, doi:10.3390/plants11243449_

Round 1
Reviewer 1 Report
The article “Microbial Interference and its Mechanisms in the Control of Plant Viruses” is devoted to the important and acute theme of the role of plant-associated microbes in plant resistance to these dangerous pathogens. The authors provided a lot of information on different ways of interaction in this tripartite system, but there are two main negative aspects:
1) The structure of the manuscript should be seriously improved.
It can be divided on 1) Introduction, 2) Microorganisms Used in the Management of Plant Virus Diseases (In this chapter information in the text shell not repeat data in tables and in my opinion, in the text authors shell discuss practical features of PGPR using), 3) Direct Microbial Antiviral Mechanisms in Viral Disease Management (only microbial antiviral compounds - lipopeptides, RNAses of bacteria etc.), 4) PGPR-induced Resistance of plants to viral pathogens (all processes in plants, which caused by microbes - PR-proteins, ROS generation, PRAses of plants). SA and JA dependent mechanisms shell be detailed in chapter (and combined with PR-proteins)
2) The manuscript contains a lot of data which overlap with the article “Mechanisms of Plant Tolerance to RNA Viruses Induced by Plant-Growth- Promoting Microorganisms” (https://www.mdpi.com/2223-7747/8/12/575). Thus, Table 2 reduplicates Table 1 from this source (authors swapped columns and expanded abbreviations). Table 3 is defined as “The bacterial and fungal bio-agents reported against the different plant viral disease 162 management are presented in Table 3, modified from Maksimov et al. [82].”Yes, the reference Maksimov et al. [82] is “Mechanisms of Plant Tolerance to RNA Viruses Induced by Plant-Growth- Promoting Microorganisms”, but it isn’t contains data on “antiviral inducing fungal organisms”(that is “antiviral inducing fungal organisms”? Authors must be more accurate with using terms).
Some paragraphs contain data, structurally and logically reduplicate the article of Maksimov et al. [82], and make reference to identical literature data (lines 289-304, 306-312, 420-427 etc). A lot of data, described in Maksimov et al. [82],is presented in the manuscript in re-written sentences.
This data is already systematized in 2019, and now a great amount of new information on plant-bacteria-virus interaction is available.
Authors can modify Table 2, if they can interpret this data in another way and diminish previously discussed data.
a few other comments:
Title: in the manuscript “microbial interference” (cross-reactions between some microorganisms) is not discussed. It should be rewritten on, for example, Mechanisms of Microbial Plant Protection and Control of Plant Viruses.
It is necessary to carry out unification of used termins (are you speaking about PGPR (only rhizobacteria) or endophytes (only existing into tissues) or plant-associated microorganisms (whole microbiome)?)
It is necessary to carefully draw up the text and correct language mistakes and incorrect use of thermins.
Author Response
|
Reviewer: 1 |
|
|
Reviewer 1 comments |
Action taken |
|
1) The structure of the manuscript should be seriously improved. It can be divided on 1) Introduction, 2) Microorganisms Used in the Management of Plant Virus Diseases (In this chapter information in the text shell not repeat data in tables and in my opinion, in the text authors shell discuss practical features of PGPR using), 3) Direct Microbial Antiviral Mechanisms in Viral Disease Management (only microbial antiviral compounds - lipopeptides, RNAses of bacteria etc.) 4) PGPR-induced Resistance of plants to viral pathogens (all processes in plants, which caused by microbes - PR-proteins, ROS generation, PRAses of plants). SA and JA dependent mechanisms shell be detailed in chapter (and combined with PR-proteins) |
· As per the reviwer suggestions, information present in the table is deleted in the text, combined the PGPR, PR-proteins and ROS generation information under ‘PGPR-induced Resistance of plants to viral pathogens’ sub headings · Modified the text in lipopeptides, RNAses of bacteria sub headings
|
|
2) The manuscript contains a lot of data which overlap with the article “Mechanisms of Plant Tolerance to RNA Viruses Induced by Plant-Growth- Promoting Microorganisms” (https://www.mdpi.com/2223-7747/8/12/575). Thus, Table 2 reduplicates Table 1 from this source (authors swapped columns and expanded abbreviations). Table 3 is defined as “The bacterial and fungal bio-agents reported against the different plant viral disease 162 management are presented in Table 3, modified from Maksimov et al. [82].”Yes (that is “antiviral inducing fungal organisms”? Authors must be more accurate with using terms). |
· The overlapped text and data has been deleted and modified according to the requirement of the article and duplicate · The reference Maksimov et al. [82] “Mechanisms of Plant Tolerance to RNA Viruses Induced by Plant-Growth- Promoting Microorganisms”, but it isn’t contains data on “antiviral inducing fungal organisms” has corrected accurately as per the reviewer comments |
|
Some paragraphs contain data, structurally and logically reduplicate the article of Maksimov et al. [82], and make reference to identical literature data (lines 289-304, 306-312, 420-427 etc). A lot of data, described in Maksimov et al. [82] is presented in the manuscript in re-written sentences. This data is already systematized in 2019, and now a great amount of new information on plant-bacteria-virus interaction is available. |
· Deleted the duplicated content and also modified the text wherever it needs revision as per the reviewer suggestions and added content in relevant place. |
|
Authors can modify Table 2, if they can interpret this data in another way and diminish previously discussed data. |
· Deleted table 1 and modified the table 2 as 1 and table 3 as 2. The table 2 has been modified and deleted the data discussed in the text |
|
a few other comments: Title: in the manuscript “microbial interference” (cross-reactions between some microorganisms) is not discussed. It should be rewritten on, for example, Mechanisms of Microbial Plant Protection and Control of Plant Viruses. |
· Title of the manuscript has been changed to ‘Mechanisms of Microbial Plant Protection and Control of Plant Viruses’ as per the reviewer suggestions |
|
It is necessary to carry out unification of used termins (are you speaking about PGPR (only rhizobacteria) or endophytes (only existing into tissues) or plant-associated microorganisms (whole microbiome)?) |
· All endophytes may not be PGPRs and we have used the terms mentioned in the research article by the researchers |
|
It is necessary to carefully draw up the text and correct language mistakes and incorrect use of thermins. |
· Taken care of all the mistakes and incorrect use of terms in the manuscript |

Reviewer 2 Report
This is a very comprehensive review of controlling plant viruses using microbial resources. Many unconventional methods are presently being employed to prevent the viral infection. Though controlling viruses is difficult, actinomycetes, bacteria, fungi and virus have been utilized for plant virus control. In addition, their possible mechanisms for suppressing viral infection in plants viacross-protection are discussed well. The number of references is also a lot, suggesting that the authors have a good understanding of literatures.
The manuscript is informative and very good.
Line 110, "pseudomonas" should be Pseudomonas.
PGPR's full name had better be provided.
Line 317, "by" is redundant.
Line 290, "Rhodopseudomonas" should be italic.
Line 521, Latine names should be double checked.
Some formats of Latin names should be double checked.
Author Response
|
Reviewer: 2 |
|
|
Reviewer 2’ comments |
Action taken |
|
Line 110, "pseudomonas" should be Pseudomonas. |
Corrected as Pseudomonas and italicized |
|
PGPR's full name had better be provided. |
The full name of PGPR’s has provided only once as ‘Plant growth promoting rhizobacteria’s’ in the text |
|
Line 317, "by" is redundant. |
Deleted the redundant word ‘by’ |
|
Line 290, "Rhodopseudomonas" should be italic. . |
The word ‘Rhodopseudomonas’ has been italicized in the text |
|
Line 521, Latine names should be double checked |
The Latine names in the manuscript has been checked many times and |
|
Some formats of Latin names should be double checked. |
The Latine names has been double checked in the entire manuscript and italicized |
|
Over all comment by the reviewer 2 This is a very comprehensive review of controlling plant viruses using microbial resources. Many unconventional methods are presently being employed to prevent the viral infection. Though controlling viruses is difficult, actinomycetes, bacteria, fungi and virus have been utilized for plant virus control. In addition, their possible mechanisms for suppressing viral infection in plants via cross-protection are discussed well. The number of references is also a lot, suggesting that the authors have a good understanding of literatures. The manuscript is informative and very good. |
|

Round 2
Reviewer 1 Report
Dear authors, I strictly recommend to you to replace data in table 1, which were copied from Maksimov et al., 2020, with the sentence, for example, "Data on bacteria/Bacillus antiviral potencial was assumed in Maksimov et al, 2020" and provide only new data on this topic in the table. Otherwise you can get the blame in plagiarism (taken together with the paragraph on RNAses, which you rewritten from this sourse by ahother words).
You can delete paragraph on RNAses or provide new literature.
You must do a serious work on plagiarism deletion and discuss your own ideas.
You must correct to a dot species names (I found Agrocyb, Enterobactera etc) and grammatics, cursive.
Author Response
|
Reviewer: 1 on 31-10-2022 |
|
|
Reviewer 1 comments |
Action taken |
|
Dear authors, I strictly recommend to you to replace data in table 1, which were copied from Maksimov et al., 2020, with the sentence, for example, "Data on bacteria/Bacillus antiviral potential was assumed in Maksimov et al, 2020" and provide only new data on this topic in the table. Otherwise you can get the blame in plagiarism (taken together with the paragraph on RNAses, which you rewritten from this source by another words). |
The overlapped table has been removed from the manuscript and it has been depicted in the text form for better clarity and to avoid duplication. |
|
You can delete paragraph on RNAses or provide new literature. |
As suggested the paragraph on RNAses has been revised with available recent literature as reflected in line no 462 to 512 |
|
You must do a serious work on plagiarism deletion and discuss your own ideas. |
Revised manuscript checked for plagiarism and conceptualized in our own ideas all along the manuscript |
|
You must correct to a dot species names (I found Agrocyb, Enterobactera etc) and grammatics, cursive. |
Corrected the dot species name of Agrocyb as Agrocybe aegerita, Enterobactera as Enterobacter aasburiae |

Round 3
Reviewer 1 Report
Dear authors,
The manuscript should be improved.
References must be in the order of citation in the text and must be placed in [ ]. I found [255] between [147] and [148], [174-189] after [204] etc. It must be improved.
Latin species names must be accurately checked. I found a lot of regular style of latin names, capital letters (like B. Subtilis), italic names of viruses (Zucchini yellow mosaic virus is italics) etc
In table 2 (and other) all host names should be “Maize, Zea mays L.” or “Zea mays L.” (trivial and binominal name or only binominal name with abbreviation of who first named).
Use abbreviation of viruses after the first mention. For example, “Cucumber mosaic virus (CMV)” is used 6-7 times.
Author Response
|
Reviewer comments |
Action taken |
|
Comments and Suggestions for Authors Dear authors,
The manuscript should be improved. |
Manuscript has been improved as per the suggestions of the reviewer |
|
References must be in the order of citation in the text and must be placed in [ ]. I found [255] between [147] and [148], [174-189] after [204] etc. It must be improved.
|
References have been arranged in order as per the suggestions of the reviewer |
|
Latin species names must be accurately checked. I found a lot of regular style of latin names, capital letters (like B. Subtilis), italic names of viruses (Zucchini yellow mosaic virus is italics) etc
|
Latin species names have been checked and italicized all the latin species names and scientific names |
|
In table 2 (and other) all host names should be “Maize, Zea mays L.” or “Zea mays L.” (trivial and binominal name or only binominal name with abbreviation of who first named).
|
In table 2, uniformity has been maintained in quoting scientific names of the crop or microorganisms |
|
Use abbreviation of viruses after the first mention. For example, “Cucumber mosaic virus (CMV)” is used 6-7 times. |
Followed in abbreviating all the virus names as per the suggestions of the reviewer |
